# A Study on the Correlation between Ship Movement Characteristics and Ice Conditions in Polar Waters

Liang Chen [1] , Changhai Huang [1,*] and Yanhao Wang [2]

1   Merchant Marine College, Shanghai Maritime University, Shanghai 201306, China; chenliang@shmtu.edu.cn
2   China Construction Harbour and Channel Engineering Bureau Group Co., Ltd., Shanghai 200433, China
*   Correspondence: chhuang@shmtu.edu.cn

**Abstract:** The opening of arctic routes provides a new option for international navigation ships. The correlation between ship movement characteristics and ice conditions should be known, which will help ships adapt to the polar waters. Based on the voyage data and sea ice manual observation data of the 'XUE LONG' ship's six voyages in polar waters, a correlation analysis model of ice conditions and ship movement characteristics was established in this work. First, the ship movement characteristics in polar waters were analyzed, such as the distribution characteristics of ship speeds, courses, and variation characteristics by using the descriptive statistical analysis method and data visualization analysis method. Then, by using multivariate correlation analysis and univariate controlled correlation analysis methods, the correlation between movement characteristics and ice conditions, such as ice concentration and thickness, and the correlation between different ice conditions themselves, were quantitatively analyzed. The result shows that the correlation analysis model of ice conditions and ship movement characteristics is reliable and effective and can obtain quantitative correlation analysis results. On the one hand, sea ice thickness has almost no significant correlation with ship movement characteristics, excluding the influence of sea ice concentration. On the other hand, excluding the influence of sea ice thickness, sea ice concentration is still significantly correlated with the absolute value of speed, speed variation, and course variation. The conclusions of this work have important reference significance for polar scientific investigations, commercial ships' voyages in icy waters, and ships' designs for icy waters.

**Keywords:** ship movement characteristics; polar waters; correlation analysis; polar navigation

## 1. Introduction

As the global climate warms, the melting of the Arctic sea ice has accelerated, which makes the Arctic waterways gradually navigable. The opening of the Arctic route will provide Asia with two more convenient routes to Europe and North America, reducing the voyage by 10–13 days [1] and saving many ship fuel costs, canal costs, security costs, personnel costs, and many more. The advantages of the Arctic route will inevitably attract a large number of merchant ships to choose it [2]. However, the current navigation environment of the Arctic waterway is still dominated by icy waters. Merchant ships choosing Arctic routes need to improve and optimize ship design, route planning, and ship maneuvering [3].

The ship movement characteristics in icy waters can provide reference information for ship design, route planning, and ship maneuvering [4]. Different sea ice conditions have different effects on ship maneuverability, ship speed, and other characteristics [5]. Due to the lack of experience in the Arctic route, there are few studies on the ship movement characteristics in icy water, and there is a lack of quantitative methods to clarify the influence of sea ice conditions on the ship movement characteristics under multiple observed variables. Therefore, a correlation analysis model of ice conditions and ship movement characteristics is established in this work based on the voyage data and sea ice manual observation data of

the 'XUE LONG' ship's six voyages in polar waters. The ship movement characteristics in polar waters are analyzed, such as the distribution characteristics of ship speeds, courses, and variation characteristics using the method of descriptive statistical analysis and data visualization analysis. The correlation of ship movement characteristics and ice conditions, such as ice concentration and thickness, and the correlation between different ice conditions themselves, are quantitatively analyzed by using multivariate correlation analysis and univariate controlled correlation analysis methods.

The remainder of this study is organized as follows. In Section 2, a literature review is presented. Section 3 presents the materials and methods. The discussion is provided in Section 4. Section 5 states the conclusion.

## 2. Literature Review

Sea ice is an important factor that affects ship movement characteristics in polar waters. The Northern Sea Route (NSR) has a large difference in ice conditions in summer and winter [6]. To ensure navigational safety, there is widespread consensus to reduce the speed of ships navigating icy waters. With the continuous improvement of the feasibility of the NSR, how to improve the efficiency and safety of ship navigation in the sea ice area has become a focus of scholars. Firstly, the relationship between ship speed and ice thickness is studied by numerical and simulation methods [7,8]. The second approach is to study the navigation performance of ships in polar open waters. Chen, C. et al., analyzed the effects of ship type, ship speed, and wave steepness on added resistance of the polar research vessel based on the three-dimensional full nonlinear time domain potential flow theory [9]. In the process of ship operation in the polar sea, some scholars have established the model of key parameters in the process of ship–ice interaction and applied it to the structural design loads [10,11]. The methods of numerical simulation and theoretical analysis provide a theoretical basis for the analysis of the ship movement characteristics in the polar sea, but it still needs to be combined with the actual observation data of the navigation environment and state for further research [12].

With the development of information technology, it has become easier to obtain Automatic Identification System (AIS) data and navigation environment data. The research results of ship movement characteristics based on a statistical analysis of data are developing continuously, but there is still a lot of potential value for massive spatiotemporal data that has not been fully utilized. The correlation between massive spatiotemporal data and ship movement characteristics needs to be deeply analyzed [13]. In terms of the characteristic state of ship movement, Zheng J. et al., studied the nonlinear characteristics of ship movement using control theory [14]. Montewka J. et al., established a hybrid model of ship performance in ice-covered waters, which can be used in ice region navigation planning [15]. With the AIS and navigation environment data, the ship movement characteristics can be further explored. Goerlandt F. et al., showed summary statistics of speed in convoy and escort operations in icy waters under different sea ice thickness situations primarily based on AIS data [16]. Zhang C. et al., pointed out that while the ice becomes denser in real life Arctic navigation, ship speed will gradually decrease [17]. The analysis of navigation environment data can also be extended to the study of traffic state characteristics [18–20].

At present, some conclusions about ship movement characteristics have been obtained from ship navigation status data and navigation environment data, but there are few studies on the quantitative correlation between sea ice density and ship movement characteristics in polar waters. Correlation research is an effective method to reveal the correlation between external variables and the target itself [21]. Zhao P. et al., proposed an analytical framework for exploring the relationship between intra-urban logistics and urban transport planning by integrating spatial analysis, network analysis, and spatial interaction analysis [22]. Gagic R. et al., determined the correlation between cruise ship activities in ports with an ambient concentration of pollutants [23]. The correlation analysis can also be extended to reveal the relationships among water traffic factors such as environmental conditions, traffic characteristics, ship collision frequency, ship defects, and so on [24,25]. In terms of polar

ship navigation, Zambon A. et al., identified the correlation between ice-induced propeller loads and sea ice conditions by experimental measurements and numerical analysis [26]. Yuen P.C. et al., analyzed the correlation of cargo damage risks for the planning of marine container transportation voyages [27]. The premise of these studies is to accurately obtain the analytical data of each target. For ships in the polar sea, how to effectively obtain the environmental state data and ship movement state data is also the focus of the research [28–31].

There are some results on the relationship between ship performance and ice conditions. In the study by Montewka J. et al., the correlation coefficient between ship speed and sea ice concentration is given; there are no detailed studies on the correlation coefficient between ship speed and different ice concentrations [32]. In fact, the sea ice concentration and sea ice thickness are also related; they are not independent of each other [33]. Kimmritz M. et al., analyzed the relationship between sea ice concentration and thickness in the sea ice data assimilation method [34]. It is assumed that sea ice concentration and sea ice thickness are both related to ship movement characteristics. In that case, it is necessary to clarify which factor is more decisive: the characteristics of the ship course or ship course variation. The relationship between the ship course and sea ice conditions also needs to be further studied. These studies can clarify whether the ship uses ice breaking or bypasses the ice when sailing in different ice regions and chooses waters with thin ice layers or waters without ice. At the same time, the ship movement characteristics of ships are not only associated with sea ice concentration but are additionally related to sea ice thickness. Then, the relationship between sea ice concentration and sea ice thickness also needs to be explored. Moreover, while sea ice concentration and sea ice thickness are each associated with ship movement characteristics, it is essential to make clear which element is more decisive.

## 3. Materials and Methods

### 3.1. Correlation Analysis Framework

The correlation analysis between ship movement characteristics and ice conditions should meet three requirements: (1) analyze the basic information of ship movement characteristics, (2) explore the interaction between ice conditions variables, (3) obtain the interaction between ship movement characteristics and ice conditions, and design a comprehensive correlation analysis framework on this basis. The framework is mainly composed of three parts: data source, core analysis method, and analysis result output, as shown in Figure 1.

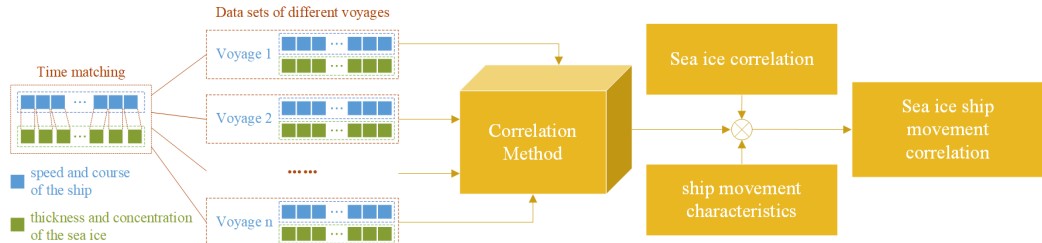

**Figure 1.** Framework for analysis of the correlation between ice conditions and ship movement characteristics.

In terms of data sources, the voyage data of the scientific research ship 'XUE LONG' in the Arctic and Antarctic, as well as the ice data collected during the voyage, will be used. Because the ice data collected by the ship during its voyage contained the actual information, they have great value for analyzing the navigation characteristics of ships in polar waters. In terms of core analysis methods, this paper studies a correlation analysis method between ice conditions and ship movement characteristics, which can output the correlation of the navigation environment, as well as the correlation between navigation environment and ship movement characteristics. In the aspect of result output, the ship navigation process is taken as the time series, the data types of the analysis results are combined, and a two-dimensional cross relationship of the correlation of data points is

formed to form a graph, and the mutual relationship between the ice conditions and the ship movement characteristics is visualized.

### 3.2. Data Collection

This study's original data came from the 5–9th Arctic voyage and the 34th Antarctic voyage of the icebreaker 'XUE LONG,' including 69,098 voyage data points from GPS equipment and 4085 manual observation data points on sea ice. The icebreaker 'XUE LONG' is China's largest polar science research ship. It is 167 m long and 22.6 m wide, with a full draft of 9 m and a gross tonnage of 15,352 tons. It has a maximum speed of 17.9 knots, an endurance of 20,000 nautical miles, and is capable of continuously breaking through 1.2 m of ice (including 0.2 m of snow) at a speed of 1.5 knots.

### 3.3. Data Preprocessing

The voyage data and the manual observation data on sea ice were matched in time. Among these two types of data, there are 836 records with the same recording time, 2435 records with a time difference of less than 1 min, and 3394 records with a time difference of less than 10 min. Considering the ship's speed and sea ice variations, 3394 records with a time difference within 10 min between the two types of data were analyzed.

(1) Sea ice concentration and its classification and measurement

Sea ice concentration represents the sea ice coverage area ratio to the total area of the sea area. It is often expressed in percentages (tenths).

In this study, the sea ice concentration data are manual observation data, and the observation interval is about 30 min.

Among them, $C_t$ represents the total concentration of sea ice in the view field with a diameter of 5 km (in tenths). $C_a$ represents the concentration of sea ice with the largest concentration in the view field with a diameter of 5 km (Type A). $C_b$ represents the concentration of sea ice with the second largest concentration in the view field with a diameter of 5 km (Type B). $C_c$ represents sea ice concentration with the third largest concentration in the view field with a diameter of 5 km (Type C).

(2) Sea ice thickness and its classification and measurement

Sea ice thickness is defined as the vertical distance between the sea ice surface and the ice bottom, in cm.

In this study, the sea ice thickness data are also manual observation data, and the observation interval is about 30 min. In the actual sea ice observation data, the interval is usually used to express sea ice thickness, such as 50–70 cm. In this study, the interval data are processed, and their values are the average values of the upper and lower boundaries of the interval.

Among them, $Th_a$ represents the thickness of the Type A sea ice. $Th_b$ represents the thickness of the Type B sea ice. $Th_c$ represents the thickness of the Type C sea ice.

$Th_{AVG}$ represents the total average thickness of sea ice in the view field with a diameter of 5 km. $Th_{AVG}$ is not directly derived from observational data and is calculated according to Formula (1) in this study.

$$Th_{AVG} = (C_a \cdot Th_a + C_b \cdot Th_b + C_c \cdot Th_c) / (C_a + C_b + C_c) \qquad (1)$$

### 3.4. Data General Characteristics and Visualization

This study performed descriptive statistical analysis on the preprocessed data to obtain ship movement characteristic information, such as ship speed and course, in the research waters. To convey and communicate information clearly and effectively with the aid of graphical means, this study conducted a data visualization analysis on the collected data to visually display the data distribution characteristics of ship movement characteristics. Histograms, rose graphs, box graphs, and so on were adopted for data visualization analysis of ship movement characteristics and ice conditions.

*3.5. Modeling*

In order to explore the quantitative correlation between sea ice conditions and ship movement characteristics, this study established a correlation analysis model of ice conditions and ship movement characteristics. The model took ice condition and ship movement state variables as inputs and included two parts: multivariate correlation analysis and univariate controlled correlation analysis, which is shown in Figure 2. In multivariate correlation analysis, the correlation between ship movement characteristics and sea ice thickness and sea ice concentration, and other state variables is mainly studied. Since the relationship between multivariate variables is very complex, ship movement characteristics may be affected by multiple variables; therefore, univariate controlled correlation analysis is established on the basis of multivariate correlation analysis, which is partial correlation analysis. By eliminating the influence of the third variable, the correlation between other variables is analyzed to obtain a more accurate relationship between ice conditions and ship movement characteristics.

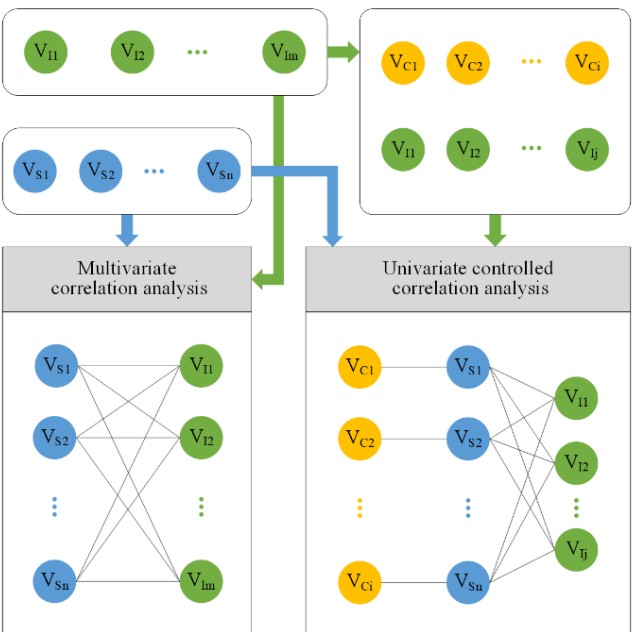

**Figure 2.** Model of correlation between ice conditions and ship movement characteristics.

As shown in Figure 2, the mathematical relationship of correlation analysis between ice conditions and ship movement characteristics is expressed as follows:

$$C_{Mi,j} = cov(V_{Si}, V_{Ij}), i \in \{1, \cdots, N\}, j \in \{1, \cdots, M\} \tag{2}$$

$$C_{UMi,j} = cov(V_{Si}, V_{Ij})|V_{Ck}, i \in \{1, \cdots, N\}, j \in \{1, \cdots, M-1\}, k \in \{1, \cdots, M\} \tag{3}$$

where $C\_M_{i,j}$ is multivariate correlation analysis, $C\_UM_{i,j}$ is the univariate controlled correlation analysis, $cov()$ is the correlation analysis function, $V_{Si}$ is the ship movement state variable, $N$ is the total number of ship movement state variables, $V_{Ij}$ is the ice condition state variable, $M$ is the total number of ice condition state variables, and $V_{Ck}$ is the selected control variable in ice condition state variables.

3.5.1. Multivariate Correlation Analysis

There are three types of data analysis methods in multivariate correlation analyses: Pearson, Spearman, and Kendall [35,36]. For quantitative variables with a normal distribution, the Pearson correlation coefficient can be used. If the data do not follow a

normal distribution or have a sorted category, the Spearman coefficient or Kendall's tau-b coefficient can be adopted [37]. The latter two measure the correlation between ranks.

Spearman's correlation is also known as rank correlation, which is used when one or both variables are ranks or ordinal scales. It is applicable to determine the degree of correlation between two variables in the case of ordinal data.

The Spearman rank correlation coefficient is defined by:

$$r_s = 1 - \frac{6\sum d_i^2}{n(n^2 - 1)} \tag{4}$$

where $d_i$ is the difference among ranks of $i$th pair of the two variables and $n$ is the number of pairs of observations.

### 3.5.2. Univariate Controlled Correlation Analysis

The univariate controlled correlation analysis calculates the partial correlation coefficient, which describes the linear relationship between two variables while controlling the effect of one or more additional variables. It can be used to determine if the relationship between two variables is direct, spurious, or intervening, controlling each of these variables' correlation with a third related variable.

The formula for the partial correlation coefficient for $X$ and $Y$, controlling for $Z$, is as follows:

$$r_{yx.z} = \frac{r_{yx} - r_{yz}r_{xz}}{\sqrt{\left(1 - r_{yz}^2\right)\left(1 - r_{xz}^2\right)}} \tag{5}$$

$r_{yx.z}$ is the partial correlation coefficient between variable $X$ and variable $Y$, controlling for $Z$.

Before solving the above formula, the zero-order coefficients between all possible pairs of variables ($Y$ and $X$, $Y$ and $Z$, $X$, and $Z$) must be calculated first. The formula for the zero-order coefficients is as follows:

$$r_{xy} = \frac{\sum_{i=1}^{n}\left[(x_i - \overline{x})(y_i - \overline{y})\right]}{\sqrt{\sum_{i=1}^{n}(x_i - \overline{x})^2 \cdot \sum_{i=1}^{n}(y_i - \overline{y})^2}} \tag{6}$$

$r_{xy}$ is the zero-order coefficient between variable $X$ and variable $Y$.

The following ranges are used for the interpretation of the strength of the correlation. Complete correlation: correlation with $|r| = 1$; high correlation or strong correlation: correlation with $0.7 \leq |r| < 1$; moderate correlation: correlation with $0.4 \leq |r| < 0.7$; low correlation or weak correlation: $|r| < 0.4$ correlation; and zero correlation: $r = 0$.

## 4. Results

### 4.1. Analysis of Ship Movement Characteristics in Polar Waters

4.1.1. Characteristics of Ship Speed in Polar Waters

The speed histogram of the "XUE LONG" ship (at intervals of 1 knot) is shown in Figure 3, and the speed statistics table is shown in Table 1.

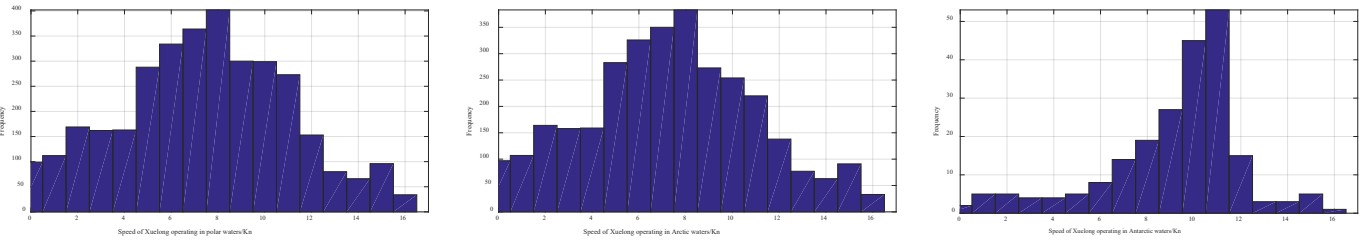

**Figure 3.** The speed distribution of the ship movement in polar waters.

**Table 1.** Statistics of ship speeds in polar waters.

|  | Minimum | Maximum | Mean | S.D. | Median | Mode | Count |
|---|---|---|---|---|---|---|---|
| Antarctic waters | 0.20 | 15.60 | 9.20 | 2.98 | 9.90 | 10.90 | 218 |
| Arctic waters | 0.00 | 16.00 | 7.38 | 3.65 | 7.40 | 8.30 | 3176 |
| Polar waters | 0.00 | 16.00 | 7.50 | 3.63 | 7.60 | 8.30 | 3394 |

From the data in the statistical table, the average ship speed in the Antarctic waters is nearly 2 knots higher than that in the Arctic waters, and ship speed variation in the Arctic waters is larger than that in the Antarctic waters.

4.1.2. Characteristics of Variation of Ship Speed in Polar Waters

In this study, the speed variation refers to the difference between the speed of the current observation time and the speed of the last observation time, and its value is a vector. The box diagram of the speed variation of the "XUE LONG" ship is shown in Figure 4, and the statistics table of speed variation is shown in Table 2. It can be seen from the box chart that the quartiles of speed variation for different waters are relatively close.

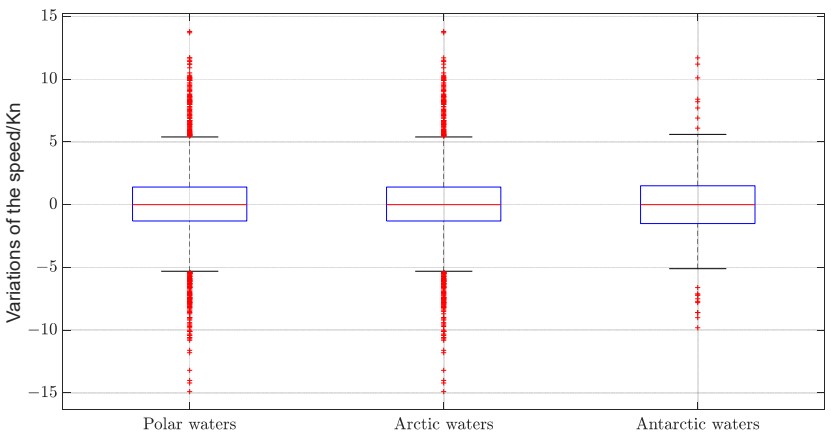

**Figure 4.** Box plot of ship speed variation in polar waters.

**Table 2.** Statistics of ship speed variation in polar waters.

|  | Minimum | Maximum | Mean | S.D. | Median | Mode | Count |
|---|---|---|---|---|---|---|---|
| Antarctic waters | −9.80 | 11.70 | −0.02 | 3.21 | 0.00 | −0.20 | 218 |
| Arctic waters | −14.90 | 13.80 | 0.00 | 2.91 | 0.00 | 0.00 | 3176 |
| Polar waters | −14.90 | 13.80 | 0.00 | 2.93 | 0.00 | 0.00 | 3394 |

From the statistical table, the mean of the speed variation in Antarctic waters and Arctic waters basically tends to be 0; the mode of speed variation in Arctic waters is 0, indicating that a certain speed is always maintained, and the mode of speed variation in Antarctic waters is −0.2 knots, which may be due to the frequency of observation during deceleration. The S.D. of speed variation in Antarctic waters is greater than that in Arctic waters, indicating a higher dispersion of speed variation.

4.1.3. Characteristics of Ship Course in Polar Waters

The rose chart of the course of the "XUE LONG" ship (with a 5° interval) is shown in Figure 5. It can be seen intuitively in the figure that, regardless of the Antarctic routes or the Arctic routes, the "XUE LONG" ship's course distribution is relatively scattered. Although there are several main lobes, the directions of the main lobes are still scattered.

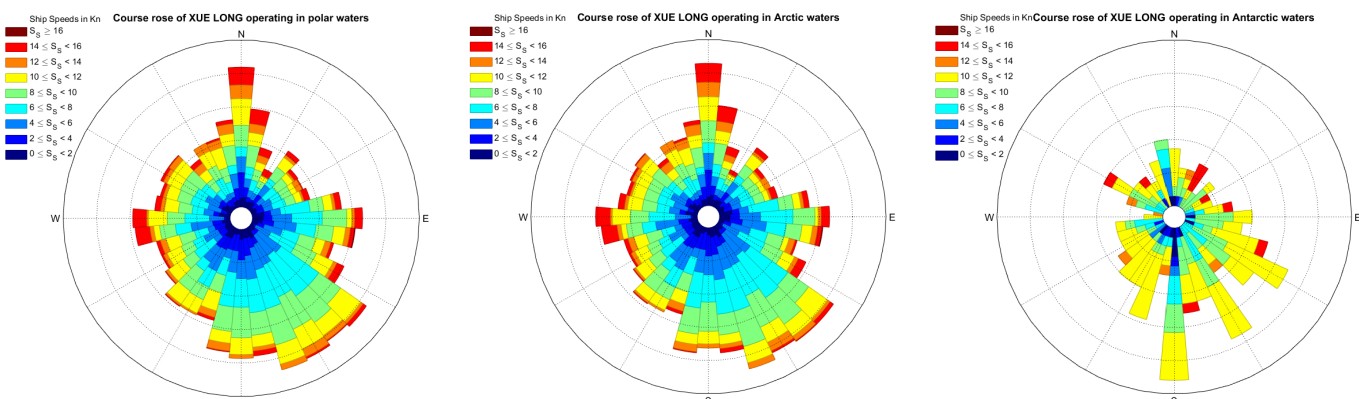

**Figure 5.** Rose chart of the ship's course distribution in polar waters.

### 4.1.4. Characteristics of the Ship's Course Variations in Polar Waters

In this study, the course variation refers to the difference between the course at the current observation time and the course at the last observation time, and its value is a vector. The box chart of the course variation of the "XUE LONG" ship is shown in Figure 6, and the statistical table of the course variation is shown in Table 3. In the box chart, it can be seen that the median of course variations in the Arctic waters is basically the same, but the interquartile range of the course variation in the Antarctic waters is larger, indicating that the course variation in the Antarctic waters is more scattered.

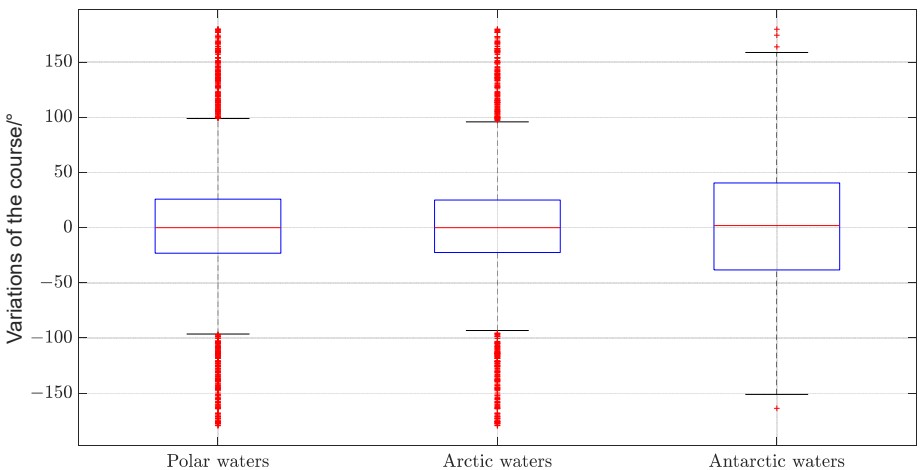

**Figure 6.** Box plot of the ship's course variations in polar waters.

**Table 3.** Statistics of the ship's course variations in polar waters.

|  | Minimum | Maximum | Mean | S.D. | Median | Mode | Count |
|---|---|---|---|---|---|---|---|
| Antarctic waters | −163.60 | 179.70 | 2.95 | 68.52 | 1.95 | −20.00 | 218 |
| Arctic waters | −179.30 | 179.80 | −0.13 | 54.62 | 0.00 | −3.20 | 3176 |
| Polar waters | −179.30 | 179.80 | 0.07 | 55.61 | 0.00 | −3.20 | 3394 |

From the statistical table, the mean of the ship's course variation in Antarctic waters deviates more from 0° than in Arctic waters, the median and mode also deviate more from 0°, and the standard deviation is larger than in Arctic waters. The ship's course variation is more scattered, and the variation is greater.

## 4.2. Correlation Analysis between Sea Ice Conditions and Ship Movement Characteristics

### 4.2.1. Correlation Analysis between Sea Ice Concentration and Ship Movement Characteristics

A bivariate correlation analysis was carried out on various types of sea ice concentrations and ship movement parameters. After testing, each variable's data does not conform to the normal distribution, so Pearson's correlation analysis cannot be used. Moreover, because the navigation parameters are not ordered variables, it is also inappropriate to use Kendall's tau-b correlation coefficient. Therefore, this study used the Spearman correlation analysis method to calculate the Spearman correlation coefficient and conduct a significance test.

The results of the correlation analysis between sea ice concentration and ship movement parameters are shown in Table 4 and Figure 7a.

**Table 4.** The correlation coefficient between sea ice concentration and ship movement characteristics.

| Spearman's Correlation Coefficient | | $C_t$ | $C_a$ | $C_b$ | $C_c$ |
|---|---|---|---|---|---|
| Ship speed | Correlation Coefficient | −0.489 ** | −0.427 ** | −0.318 ** | −0.224 ** |
| | Sig. (2-tailed) | 0 | 0 | 0 | 0 |
| | N | 3394 | 3394 | 3394 | 3394 |
| Ship speed variation | Correlation Coefficient | −0.094 ** | −0.080 ** | −0.059 ** | −0.033 |
| | Sig. (2-tailed) | 0 | 0 | 0 | 0 |
| | N | 3394 | 3394 | 3394 | 3394 |
| The absolute value of ship speed variation | Correlation Coefficient | 0.126 ** | 0.132 ** | 0.110 ** | 0.011 |
| | Sig. (2-tailed) | 0 | 0 | 0 | 0 |
| | N | 3394 | 3394 | 3394 | 3394 |
| Ship course variations | Correlation Coefficient | −0.018 | −0.007 | −0.025 | −0.009 |
| | Sig. (2-tailed) | 0.291 | 0.671 | 0.151 | 0.618 |
| | N | 3394 | 3394 | 3394 | 3394 |
| The absolute value of ship course variations | Correlation Coefficient | 0.257 ** | 0.260 ** | 0.129 ** | 0.066 ** |
| | Sig. (2-tailed) | 0 | 0 | 0 | 0 |
| | N | 3394 | 3394 | 3394 | 3394 |

**. Correlation is significant at the 0.01 level (2-tailed).

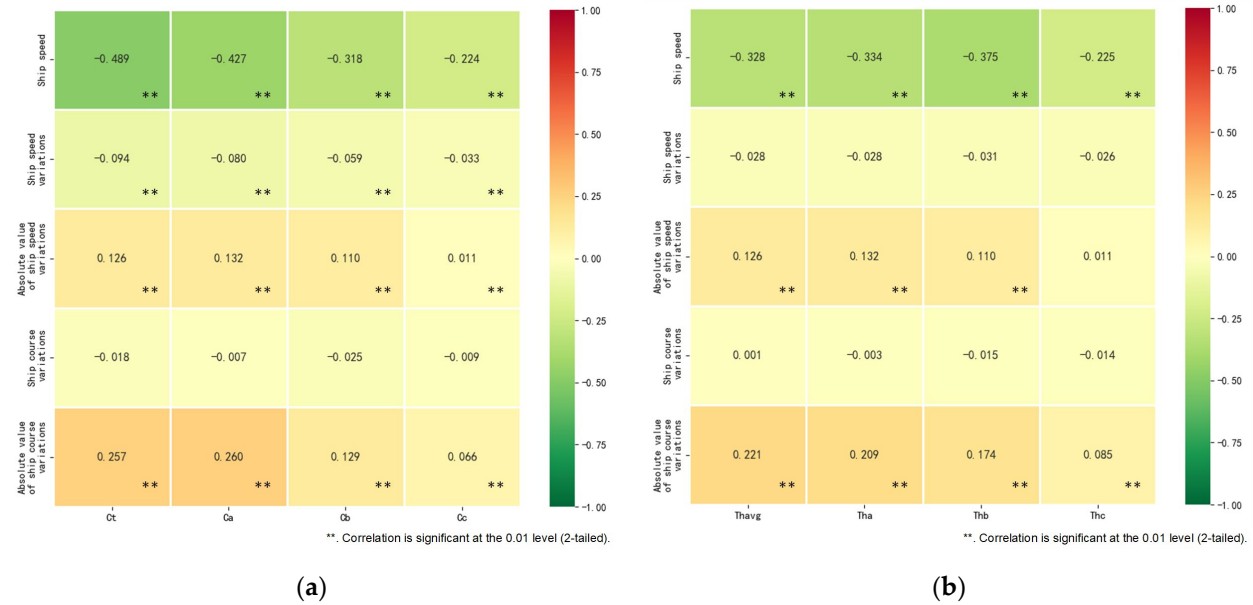

**(a)** **(b)**

**Figure 7.** The correlation coefficient. (**a**) The correlation coefficient between sea ice concentration and ship movement characteristics; (**b**) the correlation coefficient between sea ice thickness and ship speed.

It can be seen in Table 4 and Figure 7a that total sea ice concentration, Type A sea ice concentration, and ship speed have a significant and moderately negative correlation. Type B sea ice concentration, Type C sea ice concentration, and ship speed have a significant and low negative correlation. Total sea ice concentration, Type A sea ice concentration, and Type B sea ice concentration are all significantly negatively correlated with ship speed variation. Total sea ice concentration, Type A sea ice concentration, and Type B sea ice concentration are all also significantly positively correlated with the absolute value of ship speed variation. The absolute value of the correlation coefficient with the latter is greater than with the former, indicating that the correlation degree between total sea ice concentration, Type A sea ice concentration, Type B sea ice concentration, and the absolute value of the speed variation is greater than the correlation with the speed variation. There is no significant correlation between total sea ice concentration and ship course variation. Still, all sea ice concentrations have a significant weak positive correlation with the absolute value of ship course variation.

### 4.2.2. Correlation Analysis between Sea Ice Thickness and Ship Movement Characteristics

The correlation analysis results between sea ice thickness and ship movement characteristics are shown in Table 5 and Figure 7b.

**Table 5.** The correlation coefficient between sea ice thickness and ship speed.

| Spearman's Correlation Coefficient | | $Th_{AVG}$ | $Th_a$ | $Th_b$ | $Th_c$ |
|---|---|---|---|---|---|
| Ship speed | Correlation Coefficient | −0.328 ** | −0.334 ** | −0.375 ** | −0.225 ** |
| | Sig. (2-tailed) | 0 | 0 | 0 | 0 |
| | N | 3394 | 3394 | 3394 | 3394 |
| Ship speed variation | Correlation Coefficient | −0.028 | −0.028 | −0.031 | −0.026 |
| | Sig. (2-tailed) | 0.104 | 0.102 | 0.074 | 0.131 |
| | N | 3394 | 3394 | 3394 | 3394 |
| The absolute value of ship speed variation | Correlation Coefficient | 0.126 ** | 0.132 ** | 0.110 ** | 0.011 |
| | Sig. (2-tailed) | 0 | 0 | 0 | 0.516 |
| | N | 3394 | 3394 | 3394 | 3394 |
| Ship course variations | Correlation Coefficient | 0.001 | −0.003 | −0.015 | −0.014 |
| | Sig. (2-tailed) | 0.976 | 0.858 | 0.391 | 0.407 |
| | N | 3394 | 3394 | 3394 | 3394 |
| The absolute value of ship course variations | Correlation Coefficient | 0.221 ** | 0.209 ** | 0.174 ** | 0.085 ** |
| | Sig. (2-tailed) | 0 | 0 | 0 | 0 |
| | N | 3394 | 3394 | 3394 | 3394 |

**. Correlation is significant at the 0.01 level (2-tailed).

It can be seen in Table 5 and Figure 7b that there is no significant correlation between the thickness of sea ice and ship speed variation. However, the average thickness of sea ice, the thickness of Type A sea ice, the thickness of Type B sea ice, and the absolute value of ship speed variation show a significant and weak positive correlation. There is no significant correlation between sea ice thickness and ship course variation. Still, each type of sea ice thickness has a significant and weak positive correlation with ship course variations' absolute value.

### 4.2.3. Correlation Analysis between Sea Ice Concentration and Sea Ice Thickness

The relationship between sea ice conditions' characterizing factors was analyzed to explore the relationship between sea ice conditions and ship movement characteristics. The bivariate correlation analysis of various types of sea ice concentration and sea ice thickness was carried out, and the analysis results are shown in Table 6 and Figure 8.

**Table 6.** The correlation coefficient between sea ice concentration and sea ice thickness.

| Spearman's Correlation Coefficient | | $Th_{AVG}$ | $Th_a$ | $Th_b$ | $Th_c$ |
|---|---|---|---|---|---|
| $C_t$ | Correlation Coefficient | 0.597 ** | 0.602 ** | 0.609 ** | 0.356 ** |
| | Sig. (2-tailed) | 0 | 0 | 0 | 0 |
| | N | 3394 | 3394 | 3394 | 3394 |
| $C_a$ | Correlation Coefficient | 0.551 ** | 0.541 ** | 0.478 ** | 0.179 ** |
| | Sig. (2-tailed) | 0 | 0 | 0 | 0 |
| | N | 3394 | 3394 | 3394 | 3394 |
| $C_b$ | Correlation Coefficient | 0.493 ** | 0.527 ** | 0.693 ** | 0.330 ** |
| | Sig. (2-tailed) | 0 | 0 | 0 | 0 |
| | N | 3394 | 3394 | 3394 | 3394 |
| $C_c$ | Correlation Coefficient | 0.301 ** | 0.292 ** | 0.465 ** | 0.932 ** |
| | Sig. (2-tailed) | 0 | 0 | 0 | 0 |
| | N | 3394 | 3394 | 3394 | 3394 |

**. Correlation is significant at the 0.01 level (2-tailed).

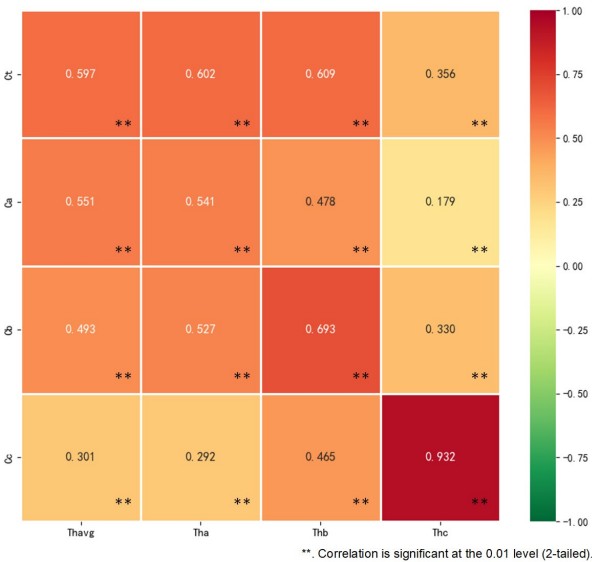

**Figure 8.** The correlation coefficient between sea ice concentration and sea ice thickness.

The total sea ice concentration is significantly positively correlated with the thickness of various sea ice types, especially with the average sea ice thickness, Type A sea ice thickness, and Type B sea ice thickness. Type A sea ice concentration has a significant and positive correlation with the thickness of various sea ice types. There is a significant and moderate positive correlation with the average sea ice thickness, Type A sea ice thickness, and Type B sea ice thickness.

Type B sea ice concentration is significantly positively correlated with the thickness of various sea ice types. The average sea ice thickness, Type A sea ice thickness, and Type B sea ice thickness are significantly positively correlated. Type C sea ice concentration has a significant and positive correlation with the thickness of various sea ice types. It has a strong positive correlation with the thickness of Type C sea ice and a significant and moderate positive correlation with Type B sea ice thickness.

### 4.2.4. Partial Correlation Analysis between Sea Ice Conditions and Ship Movement Characteristics

In view of the fact that there is a positive correlation between sea ice concentration and sea ice thickness, which characterize sea ice conditions, a partial correlation analysis method was used to analyze the relationship between sea ice conditions and ship movement

characteristics. The partial correlation analysis result is shown in Tables 7 and 8 and Figure 9a,b.

**Table 7.** Partial correlation analysis between sea ice conditions and ship movement characteristics with the control variable of ice thickness.

| Control Variable | Spearman's Correlation Coefficient | | $C_t$ | $C_a$ | $C_b$ | $C_c$ |
|---|---|---|---|---|---|---|
| $Th_{AVG}$ | Ship speed | correlation coefficient<br>Sig. (2-tailed)<br>N | −0.393 **<br>0<br>3391 | −0.333 **<br>0<br>3391 | −0.190 **<br>0<br>3391 | −0.141 **<br>0<br>3391 |
| $Th_a$ | Ship speed | correlation coefficient<br>Sig. (2-tailed)<br>N | −0.395 **<br>0<br>3391 | −0.335 **<br>0<br>3391 | −0.181 **<br>0<br>3391 | −0.148 **<br>0<br>3391 |
| $Th_b$ | Ship speed variation | correlation coefficient<br>Sig. (2-tailed)<br>N | −0.076 **<br>0<br>3391 | −0.062 **<br>0<br>3391 | −0.035 *<br>0.04<br>3391 | −0.02<br>0.254<br>3391 |
| $Th_c$ | Ship speed variation | correlation coefficient<br>Sig. (2-tailed)<br>N | −0.074 **<br>0<br>3391 | −0.060 **<br>0<br>3391 | −0.033<br>0.051<br>3391 | −0.019<br>0.263<br>3391 |
| $Th_{AVG}$ | The absolute value of ship speed variation | correlation coefficient<br>Sig. (2-tailed)<br>N | −0.006<br>0.74<br>3391 | 0.013<br>0.449<br>3391 | −0.014<br>0.411<br>3391 | −0.032<br>0.063<br>3391 |
| $Th_a$ | The absolute value of ship speed variation | correlation coefficient<br>Sig. (2-tailed)<br>N | −0.006<br>0.742<br>3391 | 0.013<br>0.436<br>3391 | −0.016<br>0.346<br>3391 | −0.031<br>0.073<br>3391 |
| $Th_{AVG}$ | Ship course variation | correlation coefficient<br>Sig. (2-tailed)<br>N | −0.031<br>0.074<br>3391 | −0.017<br>0.312<br>3391 | −0.027<br>0.114<br>3391 | −0.023<br>0.176<br>3391 |
| $Th_a$ | Ship course variation | correlation coefficient<br>Sig. (2-tailed)<br>N | −0.027<br>0.118<br>3391 | −0.014<br>0.409<br>3391 | −0.025<br>0.146<br>3391 | −0.022<br>0.204<br>3391 |
| $Th_{AVG}$ | The absolute value of ship speed variation | correlation coefficient<br>Sig. (2-tailed)<br>N | −0.006<br>0.74<br>3391 | 0.013<br>0.449<br>3391 | −0.014<br>0.411<br>3391 | −0.032<br>0.063<br>3391 |
| $Th_a$ | The absolute value of ship course variation | correlation coefficient<br>Sig. (2-tailed)<br>N | 0.170 **<br>0<br>3391 | 0.206 **<br>0<br>3391 | 0.006<br>0.737<br>3391 | −0.015<br>0.38<br>3391 |

**. Correlation is significant at the 0.01 level (2-tailed). *. Correlation is significant at the 0.05 level (2-tailed).

**Table 8.** Partial correlation analysis between sea ice conditions and ship movement characteristics with the control variable of ice concentration.

| Control Variable | Spearman's Correlation Coefficient | | $Th_{AVG}$ | $Th_a$ | $Th_b$ | $Th_c$ |
|---|---|---|---|---|---|---|
| $C_t$ | Ship speed | correlation coefficient<br>Sig. (2-tailed)<br>N | −0.033<br>0.057<br>3391 | −0.028<br>0.103<br>3391 | −0.042 *<br>0.015<br>3391 | −0.018<br>0.298<br>3391 |
| $C_a$ | Ship speed | correlation coefficient<br>Sig. (2-tailed)<br>N | −0.107 **<br>0<br>3391 | −0.107 **<br>0<br>3391 | −0.147 **<br>0<br>3391 | −0.130 **<br>0<br>3391 |
| $C_t$ | Ship speed variation | correlation coefficient<br>Sig. (2-tailed)<br>N | 0.026<br>0.131<br>3391 | 0.022<br>0.2<br>3391 | 0.032<br>0.064<br>3391 | 0.006<br>0.732<br>3391 |

**Table 8.** *Cont.*

| Control Variable | Spearman's Correlation Coefficient | | $Th_{AVG}$ | $Th_a$ | $Th_b$ | $Th_c$ |
|---|---|---|---|---|---|---|
| $C_a$ | Ship speed variation | correlation coefficient | 0.013 | 0.008 | 0.013 | −0.011 |
| | | Sig. (2-tailed) | 0.461 | 0.634 | 0.448 | 0.528 |
| | | N | 3391 | 3391 | 3391 | 3391 |
| $C_t$ | The absolute value of ship speed variation | correlation coefficient | 0.036 * | 0.036 * | −0.012 | 0.008 |
| | | Sig. (2-tailed) | 0.034 | 0.035 | 0.49 | 0.625 |
| | | N | 3391 | 3391 | 3391 | 3391 |
| $C_a$ | The absolute value of ship speed variation | correlation coefficient | 0.028 | 0.029 | −0.013 | 0.01 |
| | | Sig. (2-tailed) | 0.097 | 0.097 | 0.454 | 0.559 |
| | | N | 3391 | 3391 | 3391 | 3391 |
| $C_t$ | Ship course variation | correlation coefficient | 0.019 | 0.012 | 0.007 | −0.007 |
| | | Sig. (2-tailed) | 0.277 | 0.485 | 0.701 | 0.672 |
| | | N | 3391 | 3391 | 3391 | 3391 |
| $C_a$ | Ship course variation | correlation coefficient | 0.01 | 0.003 | −0.002 | −0.013 |
| | | Sig. (2-tailed) | 0.563 | 0.841 | 0.918 | 0.445 |
| | | N | 3391 | 3391 | 3391 | 3391 |
| $C_t$ | The absolute value of ship course variation | correlation coefficient | 0.050 ** | 0.041 * | −0.031 | −0.013 |
| | | Sig. (2-tailed) | 0.003 | 0.016 | 0.073 | 0.443 |
| | | N | 3391 | 3391 | 3391 | 3391 |
| $C_a$ | The absolute value of ship course variation | correlation coefficient | 0.052 ** | 0.046 ** | 0.001 | 0.032 |
| | | Sig. (2-tailed) | 0.002 | 0.007 | 0.962 | 0.059 |
| | | N | 3391 | 3391 | 3391 | 3391 |

**. Correlation is significant at the 0.01 level (2-tailed). *. Correlation is significant at the 0.05 level (2-tailed).

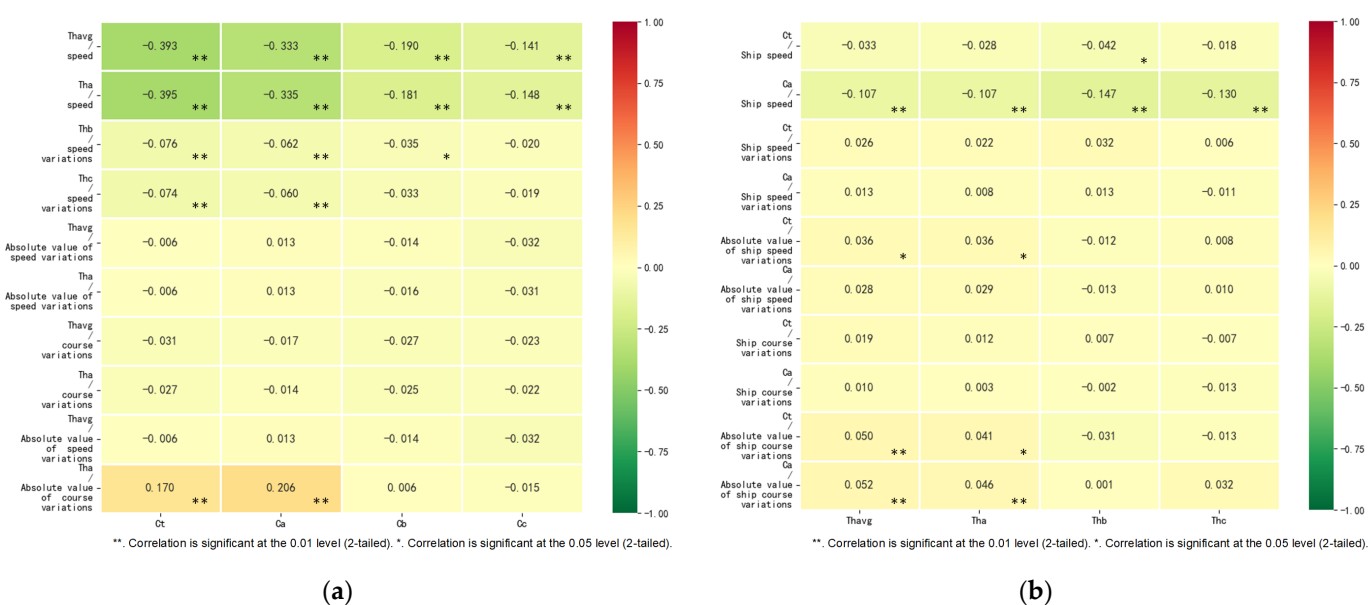

**Figure 9.** Partial correlation analysis. (**a**) Partial correlation analysis between sea ice conditions and ship movement characteristics with the control variable of ice thickness; (**b**) partial correlation analysis between sea ice conditions and ship movement characteristics with the control variable of ice concentration.

(1) Partial correlation analysis between sea ice conditions and ship speed

While controlling for $C_t$, there is almost no significant correlation between ship speed and sea ice thickness. While controlling for $C_a$, the speed and the thickness of various sea ice types show a significant and weak negative correlation. While controlling for $Th_{AVG}$, the speed and various types of sea ice concentration all show significant and weak negative

correlations. While controlling for $Th_a$, the speed and various types of sea ice concentration all show significant and weak negative correlations.

(2) Partial correlation analysis between sea ice conditions and ship speed variations

While controlling for $C_t$, there is no significant correlation between the speed variations and the sea ice thickness. While controlling for $C_a$, there is no significant correlation between the speed variations and the sea ice thickness. While controlling for $Th_{AVG}$, there are significant and weak correlations between ship speed variations and total sea ice concentration, Type A sea ice concentration, and Type B sea ice concentration, but there is no significant correlation with Type C sea ice concentration. While controlling for $Th_a$, the speed variations have significant and weak correlations with total sea ice concentration and Type A sea ice concentration but do not significantly correlate with Type B and Type C sea ice concentration.

(3) Partial correlation analysis between sea ice conditions and the absolute value of speed variations

While controlling for $C_t$, ship speed variations' absolute value has a significant and weak correlation with sea ice's average thickness and Type A sea ice thickness. Still, it does not significantly correlate with the thickness of Type B and Type C sea ice. While controlling for $C_a$, the absolute value of the speed variations does not significantly correlate with the thickness of various sea ice types. While controlling for $Th_{AVG}$, the absolute value of the speed variations does not significantly correlate with the concentration of various sea ice types. While controlling for $Th_a$, the absolute value of the speed variations does not significantly correlate with the concentration of various sea ice types.

(4) Partial correlation analysis between sea ice conditions and ship course variations

While controlling for $C_t$, there is no significant correlation between ship course variations and the thickness of various sea ice types. While controlling for $C_a$, ship course variations have no significant correlation with the thickness of various sea ice types. While controlling for $Th_{AVG}$, there is no significant correlation between ship course variations and the concentration of various sea ice types. While controlling for $Th_a$, there is no significant correlation between ship course variations and the concentration of various sea ice types.

(5) Partial correlation analysis between sea ice condition and the absolute value of ship course variations

While controlling for $C_t$, ship course variations' absolute value has a significant and weak positive correlation with sea ice's average thickness and Type A sea ice thickness. Still, it does not significantly correlate with the thickness of Type B and Type C sea ice. While controlling for $C_a$, the absolute value of ship course variations is significantly and positively correlated with the average sea ice thickness and the thickness of Type A sea ice, but not significantly correlated with the thickness of the Type B and Type C sea ice. While controlling for $Th_{AVG}$, the absolute value of ship course variations does not significantly correlate with the concentration of various sea ice types. While controlling for $Th_a$, the absolute value of ship course variations is significantly and positively correlated with sea ice concentration and Type A sea ice concentration, but not significantly correlated with Type B and Type C sea ice concentration.

## 5. Discussion

### 5.1. Ship Movement Characteristics in Polar Waters

The ship movement characteristics in icy waters are a reference value for ship design and management. This study extracts the ship movement characteristics in the Arctic and Antarctic icy waters based on statistical data.

(1) Frequent changes in speed

The speed distribution chart in Figure 3 and the speed variation box chart in Figure 4 show that the speed distribution of the "XUE LONG" ship is relatively scattered, and the speeds vary more frequently. This may be due to the narrow navigable waters between the ice, the ship's change of heading being hindered, or the ship needing to break the ice, which usually requires the ship to keep changing speed.

(2) Frequent changes in the course

From the ship's course rose chart in Figure 5 and the course variation box chart in Figure 6, it can be seen that the course distribution of the "XUE LONG" ship is relatively scattered, and the course varies more frequently. A popular explanation is that to avoid icy conditions, ships often have to change their course.

(3) Navigating at a low speed

The "XUE LONG" has a maximum speed of 17.9 knots. The average speed is lower than 10 knots in actual navigation, and the speed is relatively low. Once the ship enters an icy area, it will usually be prepared to slow down before the situation is determined.

(4) The difference exists between the Antarctic and Arctic routes

The Antarctic routes' speeds are more concentrated, most of which are distributed in the 8–12 knot range; the kurtosis of the speed distribution of the Arctic route is smaller than that of the Antarctic route, and the speeds appear more frequently in the range of 2–12 knots. The Antarctic route's average speed is nearly 2 knots higher than that of the Arctic route, and the standard deviation of the speeds of the Arctic route is large, and the speeds vary greatly. The distribution patterns of speed variations of different routes are relatively similar, but the Antarctic route's dispersion degree of speed variations is higher. Whether in the Antarctic routes or the Arctic routes, the distribution of the courses of the "XUE LONG" ship is relatively scattered, but the distribution of the course variation is more concentrated and the left and right forms of the distribution curve are closer; the course varies in the Antarctic routes because they are more scattered than the Arctic routes, and the degree of change in course variation is greater.

*5.2. The Influence of Sea Ice Conditions on Ship Movement Characteristics*

This study shows that, in general, both sea ice concentration and sea ice thickness will affect the speed and course variations of ships. Correlation analysis was carried out on different sea ice conditions and the "XUE LONG" ship's ship movement characteristics. The main conclusions are as follows.

5.2.1. The Relationship between Sea Ice Concentration and Ship Movement Characteristics

(1) There is a significant negative correlation between sea ice concentration and ship speed. This shows that the speed decreases with the increase in sea ice concentration and vice versa. In the study presented by Montewka [16], the correlation coefficient between speed and level ice concentration is $-0.610$, and the correlation coefficient between speed and ridged ice concentration is $-0.31$; the correlation coefficient between speed and total ice concentration calculated in this study is $-0.489$, which is between the above two values. The negative correlation between sea ice concentration and ship speed can also be clearly seen in the study's time series of analyzed parameters.

(2) The sea ice concentration has a significant negative correlation with speed variation. In contrast, it has a significant and weak positive correlation with the absolute value of speed variation, and the correlation degree is greater than the correlation with the speed variation. With the increase in sea ice concentration, the magnitude of speed variation increases, and the direction of speed variation can be decelerated or accelerated.

(3) There is no significant correlation between sea ice concentration and course variation. Still, the concentrations of all sea ice types have a significant and weak positive correlation with the absolute value of course variation. As the concentration of sea ice increases, the course variation's magnitude increases, but the course variation's direction is uncertain.

5.2.2. The Relationship between Sea Ice Thickness and Ship Movement Characteristics

(1) The thickness of all sea ice types has a significant and weak negative correlation with ship speed. This shows that the ship's speed decreases with the increase in sea ice thickness and vice versa. In the study by Montewka [16], the correlation coefficient between speed and level ice thickness is $-0.120$, and the correlation coefficient between speed and

rafted ice thickness is −0.500. The correlation coefficient between speed and average ice thickness calculated in this study is −0.328, which is between the above two values. The negative correlation between sea ice thickness and ship speed can also be found in the h–V curve, which describes the ship's maximum speed in level ice in the study by Izumiyama [38].

(2) There is no significant correlation between sea ice's average thickness and speed variation. However, the average thickness, Type A sea ice thickness, and Type B sea ice thickness are all positively correlated with the speed variation's absolute value; as the thickness of sea ice increases, the magnitude of speed variation increases.

(3) There is no significant correlation between sea ice thickness and course variation. Still, all sea ice thickness types have a significant and weak positive correlation with the absolute value of course variation. As the thickness of sea ice increases, the magnitude of course variation increases.

### 5.2.3. The Correlation between Sea Ice Concentration and Sea Ice Thickness

The overall sea ice concentration has a significant positive correlation with the thickness of various sea ice types. The average sea ice thickness, Type A sea ice thickness, and Type B sea ice thickness have a significant and moderate positive correlation, which can be explained by reduced horizontal melting for thicker ice.

### 5.2.4. Partial Correlation Analysis between Sea Ice Conditions and Ship Movement Characteristics

Since sea ice concentration and sea ice thickness have a significant positive correlation, a partial correlation analysis of sea ice conditions and ship movement characteristics was performed. Excluding sea ice concentration, sea ice thickness has almost no significant correlation with ship movement characteristics; excluding the influence of sea ice thickness, sea ice concentration is still significantly correlated with the absolute value of speed, speed variation, and course variation. That is to say, when the sea ice concentration is high, the speed of the ship will be reduced, and the magnitude of the course will be increased. Overall this can be explained by the fact that the ship tends to navigate in more open waters with the same ice thickness.

### 5.3. Shortcomings and Prospects

The data sampling in this study is not equal in the Antarctic and Arctic routes. The data collection and analysis need to be improved to further analyze the difference in ship movement characteristics in the Antarctic and Arctic routes. There may be errors in the observation data itself, and the data should be verified and preprocessed to improve accuracy. In addition to the sea ice concentration and sea ice thickness discussed in this study, the effects of sea ice types, etc., need to be further studied.

## 6. Conclusions

In this paper, we introduced a correlation analysis model for ice conditions and ship motion characteristics. The original work on correlation analysis was first developed since the complexity of Arctic navigation and the lack of sufficient navigational experience may make it difficult to discern the relationship between all the factors that influence the ship movement characteristics. The whole correlation analysis model consists of two sub-processes: one is multivariate correlation analysis and the other is univariate controlled correlation analysis. With regard to the former, the Spearman correlation method was chosen because it is more suitable for determining the degree of correlation between two variables in the case of ordinal data, and is applicable to the actual voyage and ice data used in this paper. According to the analysis result, the correlation coefficient between speed and total ice concentration calculated in this paper is −0.489. The sea ice concentration has a significant negative correlation with the speed variation and the concentrations of all

sea ice types have a significant and weak positive correlation with the absolute value of course variation.

Since the relationships between multivariate variables is very complex, the ship movement characteristics may be affected by more than one variable. In order to further explore the relationship between ice conditions and ship movement characteristics, a univariate controlled correlation analysis was developed based on the multivariate correlation analysis. The results show that excluding sea ice concentration, sea ice thickness has almost no significant correlation with ship movement characteristics. Excluding the influence of sea ice thickness, sea ice concentration is still significantly correlated with the absolute value of speed, speed variation, and course variation. The conclusions of this work have important reference significance for polar scientific investigations, commercial ships' voyages in icy waters, and ships' designs for icy waters. Notwithstanding this, there are certain limitations that require further effort in future research. First of all, the input parameters of the current correlation analysis model would benefit from including a greater number of related components, such as flow velocity, flow direction, wind conditions, etc., which would facilitate a more comprehensive analysis of the ship movement characteristics in polar waters. Second, since the current study is based on the "XUE LONG" ship and the data are not extensive enough, the next step is to try to obtain more data from different types of ships. Finally, the relationship between ship motion characteristics and icy conditions in polar waters is not only a static correlation analysis problem but also feedback on ship maneuvers. This means that the next research effort will combine the captain's maneuvering and decision-making processes in polar waters to provide a practical reference for more ships attempting to navigate Arctic routes.

**Author Contributions:** Conceptualization, L.C., C.H. and Y.W.; methodology, L.C. and C.H.; validation, C.H.; investigation, C.H. and Y.W.; resources, L.C. and C.H.; data curation, C.H.; writing—original draft preparation, C.H. and Y.W.; writing—review and editing, L.C. All authors have read and agreed to the published version of the manuscript.

**Funding:** This work was supported by the National Natural Science Foundation of China (Grant No. 51909156, 51709167) and the China Postdoctoral Science Foundation (Grant No. 2016M591651).

**Institutional Review Board Statement:** Not applicable.

**Informed Consent Statement:** Not applicable.

**Data Availability Statement:** 3rd Party Data. Restrictions apply to the availability of these data. Data was obtained from the "XUE LONG" ship and are available from the authors with the permission of the "XUE LONG" ship.

**Acknowledgments:** The authors acknowledge the National Arctic and Antarctic Data Center, Polar Research Institute of China for supporting the ship voyage data and manual sea ice observation data for the present work. The authors also pay high tribute to the Chinese Polar Research Team.

**Conflicts of Interest:** The authors declare no conflict of interest.

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
