# Peer review of "A Study on the Correlation between Ship Movement Characteristics and Ice Conditions in Polar Waters"

_jmse, doi:10.3390/jmse11040729_

Round 1
Reviewer 1 Report
The authors present an analysis of ship movements (speed and heading) in ice conditions, and correlate the dependent variable (ship speed) with the independent variables (ice parameters). While the analysis seems to very detailed at the statistical level, the authors do not give a lot of detail on the operating practice on board the ship. For example, are the the speed and changes voluntary or involuntary? Does the captain of the ship adjust the speed for different ice conditions? Similarly does the captain change heading to avoid ice conditions, or stay in more favourable ones?
Perhaps the authors could clarify how the data segments were analyzed. Are they for the full voyage, or for selected segments? Also, so overall specification on the ship such as dimensions, installed power, design speed, and ice class, would add some context.
Overall, the paper is well written.
Reviewer 2 Report
A paper examining an interesting topic, but suffering from lack of clarity. There is a need to rewrite your abstract (preferably from scratch, see detailed guidance in the attached pdf), as well as revisit your introductory and literature review sections so to be able to "prepare/explain" about your overall framework of research and your "intended purpose". Finally, there is a need to include "certain references" in your introductory section...Something similar applies to your conclusions section (too short/not enough and with the first few sentences fitting better in your introductory paragraph).
Good luck with your revision!

Round 2
Reviewer 2 Report
Overall appearance has improved. Restructuring the abstract was a good choice, since the reader can now easily understand "what" is attempted and the expected result.
Although I have not been through the mathematical equations (will leave this task to the editor's discretion), it is now fine to proceed towards the publishing stage.